# Comparison of Structural Analysis of Thin-Walled Structures Accomplished by Isogeometric Analysis and the Finite Element Method

**DOI:** 10.3390/ma15196516

**Published:** 2022-09-20

**Authors:** Jozef Bocko, Patrik Pleško, Ingrid Delyová, Peter Sivák

**Affiliations:** Department of Applied Mechanics and Mechanical Engineering, Faculty of Mechanical Engineering, Technical University of Košice, Letná 9, 042 00 Košice, Slovakia

**Keywords:** isogeometric analysis, thin-walled elements, finite element method, stress

## Abstract

Isogeometric analysis (IGA) represents a relatively new method of problem-solving in engineering practice. A huge advantage of this method over the finite element method (FEM), is the reduction of the simulation execution time. Non-uniform rational B-splines (NURBS) allow the use of higher-order basis functions, thus increasing the accuracy of the solution. This paper deals with the comparison of structural analysis of thin-walled structural elements using isogeometric analysis and the finite element method. The investigated objects are modelled using a single patch in MATLAB. The basic functions are created from NURBS, which were previously used in the creation of an accurate geometric model. The paper contains a comparison of the results obtained by the above-mentioned methods. All computations are performed in the elastic domain.

## 1. Introduction

A wide range of applications of IGA in the fields of linear elasticity, vibration, elastodynamics, nonlinear problems, incompressible solids and fluid mechanics are presented in [1], which can be considered a fundamental milestone of this method. Isogeometric analysis is a relatively new method for problem-solving. Since IGA uses the same basic functions in the creation of the geometry and in the expression of physical tensor quantities such as displacements, temperature, etc., it reduces the time to perform the simulation. Which is an advantage over FEM. NURBS allows the use of higher-order basis functions, thus increasing the accuracy of the solution. This is mainly due to the ability to create arbitrary shapes, including perfect circles. Of course, even with this method, one can encounter “refinement” which serves to increase the flexibility of NURBS. All this knowledge is perfectly suited to working with plates of different shapes, or with any shapes of shells. Time savings are obtained by not having to create new geometry by meshing. IGA is nowadays a very accurate method for analysing a wide range of problems.

It should be noted that there are several programs where surfaces can be created using NURBS. Mostly these are preprocessors only, with the exception of MATLAB and Rhinoceros 3D after downloading add-ons. Several of these are described in [2].

Mention should be made of the work [3], which is one of the few works that provides a comprehensive tutorial for the creation of NURBS surfaces and at the same time provides the basis for IGA of plates and shells in MATLAB. This work, like the work of [1], has been a source of knowledge for the understanding of Bézier curves and surfaces, B-splines and their special variant NURBS.

In [4], one can take a closer look at the possibility of creating a model that is composed of multiple “patches”. Additionally described are hierarchical and parametric shell models, which have different numbers of parameters depending on the desire to achieve the most accurate result. The emergence of the “locking” phenomenon is also described in detail and can be encountered in [5]. Shells and plates represent the basic structural elements. They are three-dimensional objects where one dimension is smaller than the remaining two. Due to their curvature, shells are able to carry transverse loads and allow for optimal material utilization. Shells and their behaviour have been a very common object of reflection throughout history, and many theories have been developed to describe their behaviour more or less accurately. Among the best-known shell theories are the Byrne relations, the Love–Timoshenko relations, the Reissner–Naghdi relations, etc. For boards, the Kirchhoff–Love theory, the Reissner–Mindlin theory, the Reissner–Stein theory, and many others have arisen. Plates are a special case of shells. To understand the behaviour of shells, we have relied on the literature [6,7,8]. For plates, we have focused mainly on the work of [8], as it provides a more than sufficient basis for understanding the plate problem.

When comparing FEA with IGA, it is worth pointing out the works of [9,10,11,12], which represent a very rich source of information for understanding FEA. Only then can the capabilities and properties of NURBS elements be fully appreciated.

## 2. Current Status of the Issue

Since 2005, many papers have been published on the topic of IGA. In this work, the contribution of predecessors who have used IGA for various shell, slab or volume problems cannot be avoided. Works that have used FEM and other, similar methods for shell analysis have also found a place here.

Among the interesting methods developed from IGA is the isogeometric mesh-free collocation approach. In [13] it was used for static, vibration and stability analysis of a laminate composite. It is a promising alternative to the Galerkin method. The collocation method in the isogeometric analysis is used to reduce the computational time and increase the convergence rate. In [14], the authors focused on the stabilized space–time formulation for linear and nonlinear elastodynamics. In this work, they set two objectives. The first was to introduce stabilized weak formulations in the linear and nonlinear context. The second was the use of the Galerkin scheme in space and time with higher order and higher continuity of the basis function. To express the numerical performance of this problem, this method is applied to typical shock or vibration problems that are commonly solved in elastodynamics. The current trend is to reduce the time required for the simulation, and this has not been avoided in the solution of the problems associated with the rolling wheel [15]. Current numerical methods provide an opportunity to understand this complex phenomenon. Of the many approaches to solving such a problem, FEM is the most widely used in engineering practice. IGA, in turn, provides an accurate and more efficient geometric description of the body. B-splines have been used to describe the wheel, the continuity of the geometry and the array of variables. The high degree of model continuity allows the analysis to be less sensitive to discretization. When compared with FEM, a difference can be noticed at this point.

At present, the FCP is at a very high level. It allows to solve even nonlinear problems, see [16]. The novelty in this work is a family of shell finite elements developed for geometrically nonlinear tube bending analysis. The first Piola–Kirchhoff stress tensor and the conjugate gradient of the virtual displacement field are introduced in the framework of the virtual work principle. The constitutive description is based on the Saint Venant–Kirchhoff model. More recent works include [17]. In it, a new quadrilateral element for doubly curved laminate composite shells is presented. Mindlin’s shell theory, which includes von Kármán type nonlinearities, was used in the formulation of the governing equations. The publication [18] describes the development of finite element matrices for shell and plate elements. The authors concluded that for FEM, the formulations between plates and 2D solids are very similar.

In [19], Bernoulli-Euler beams are analysed using IGA. This is an efficient and accurate numerical approach for static bending and vibration analysis. This method includes a deformation gradient, a rotation gradient and a velocity gradient, which influence simultaneously through a reformulated stress elasticity gradient with only one material parameter for each gradient influence. Based on Hamilton’s principle and IGA, the governing equations are derived and solved, satisfying the requirements for a higher degree of continuity in the microstructure-dependent Bernoulli–Euler beam.

The paper [20] deals with the simulation of fracture initiation under impact loading on a shell made of a composite composed of metallic elements using X-FEM. The fracture is described as a simple 1D mesh located on the centroidal surface and is independent of the finite element mesh. The fracture criterion is based on the measurement of the equivalent stress at the crack root. A similar problem of fracture in shells is addressed in [21]. A local form of the equilibrium laws for the continuum was expressed. These were used to express the resulting stresses. A special feature is the use of the mesh-free Galerkin formulation using differential operators.

Cracks can also be solved using IGA [22], where IGA has been used for the crack root only and a structured mesh based on the B-spline finite element method (BSFEM) is used for the remaining area. BSFEM is used in the global approximation and IGA was applied to the crack surroundings, which are covered by only a few elements. The geometry of the selected region is described using the basic functions of B-splines in the parameter space. A specific function with the relevant parameters is inserted into the geometric description to determine the type of solution around the crack shank. The basic functions of the B-splines are used as the basis functions to compute the displacement field in the local region in parametric space.

FEA and IGA are not the only options for plate and shell analysis. The mesh-free Petrov-Galerkin method can also be used to solve the shell problem in three-dimensional space. The work [23] provides a brief introduction to understanding mesh-free methods. Additionally, the work [24] deals with the analysis of plates and shells using mesh-free methods. The objects analysed are composite structures. Galerkin’s mesh-free method and Kernel particle method were used.

In [25], new basis functions for IGA using NURBS have been introduced, with the main goal of making it easier to introduce Dirichlet boundary conditions. These new basis functions were obtained by embedding rational local Lagrangian interpolations (as T-splines) and were computed in a process similar to the finite difference method. Due to the equivalence with NURBS, the new basis functions are named “non-uniform” rational Lagrange basis functions (NURLs) in this paper. Dirichlet boundary conditions can be directly embedded in the IGA using NURLs because they are interpolations of the basic functions.

The publication [7] can be considered a fundamental work in the field of plate and shell theory, but the reader may find more useful the more recent work [6], which contains detailed descriptions of the theories of plates (quadrilateral and circular) and shells. Space is devoted to shear deformations, stability and dynamic analysis. The last chapter is devoted exclusively to the analysis of plates using FEM. In [3,8,26] it was not the primary task to deal with shell theory, but chapters describing shells, plates and their analytical calculation can be found. FEM also allows modelling plates and shells that are made up of fibre-reinforced composite materials. In [27] formulations for large deformations are also described. The orientation of the fibres can be either random or purposely chosen in the analysis.

Nowadays, more and more of the professional community is dealing with composite materials, and we have already mentioned several papers that discuss shells and plates that are composite using FEA, or mesh-free methods. Sandwich shells and plates are also among the quite common materials. In [28], a special kind of element was used to describe the structural behaviour and bonding of a sandwich shell. To eliminate interlaminar stress discontinuities and to satisfy boundary conditions, a method is used to accurately compute the transverse shear and normal stresses in the laminate shell. In [29], cells that can be considered to be shells are considered. A numerical model was constructed to obtain the desired results.

Thanks to the addition of embedding to the IGA for Kirchhoff–Love shells in [30], the treatment of even large elastoplastic deformations is possible. The main goal was to reformulate the governing equations of the thin shell. Numerical examples, ranging from static to dynamic tests, demonstrate accuracy, efficiency and robustness. There are also earlier works where Kirchhoff–Love shell elements were solved using IGA, without various add-ons. For a more detailed procedure, it is worthwhile to consult the work [31]. Kirchhoff–Love shell elements also have the advantage that the necessary continuity between the elements is easily achieved. The Rhinoceros 3D software was used for the modelling.

IGA has also found its application in creating a high-quality mesh, which would be as conformal as possible, for the body shape in complex composite microstructures [32]. Micro CT photographs of composites reveal many irregularly shaped inclusions. Due to Nitsche’s method, volume connections between the inclusions and the main domain of the structure are embedded.

In modelling, it may be the case that the model consists of multiple “patches”. Sometimes it may happen that the surfaces formed from NURBS, intersect each other and need to be clipped. In this case, the IGA discontinuous Galerkin method (hereafter referred to as IGA DG) is used to unify them. Its use can be encountered in the work [33]. The surfaces are required to be C0 continuous for the IGA DG to be applicable. By extending the IGA DG, the problem of overlapping surfaces can be solved.

In [34], a discretization of Kirchhoff–Love thin shells based on the splitting algorithm is presented. Symmetry is exploited to increase the efficiency of the analysis by splitting the surfaces. This new algorithm provides an improvement in the flexibility of the current IGA, while expanding the area of possible applications.

IGA, as mentioned above, finds its application not only in homogeneous materials. The work [35] describes the use of IGA for a composite material containing graphene layers under load on a strut while varying the temperature and the distribution of the layers.

The 2012 work [36] proved that IGA is able to compete with classical FEA in a field that has very high demands on the accuracy and precision of the results, such as the aerospace industry. The deformation of the turbine blade was assumed to be linearly elastic with respect to all types of loads and boundary conditions that normally occur. This assumption was verified using classical FEM. This resulted in nearly identical outputs, but with only a fraction of the degrees of freedom for IGA versus FEM.

### 2.1. Preprocessor Overview for IGA

In the following section, a brief overview of some of the programs that allow geometry creation using NURBS is described.

#### 2.1.1. Rhinoceros 3D

Rhinoceros 3D is a specially designed program for modelling NURBS curves and surfaces. In Rhinoceros 3D, anything can be modelled as a NURBS. It uses output formats with the extension .3DM and .IGES. It is equipped with a number of methods for NURBS modelling. The geometry can be easily edited at all times. Access to NURBS is quite straightforward, it is easy to change control point positions, edit nodal vectors and rotate curves and surfaces. It is mainly designed for 2D mode-matching, but can also be applied to 3D.

#### 2.1.2. Inkscape

It is a vector graphic editor. For IGA it is usable as a simple preprocessor. It uses the .SVG format, but can be formatted to .XML. The data to describe a Bézier curve can be easily obtained.

#### 2.1.3. AutoCAD

Since AutoCAD contains NURBS curves, it is usable as a preprocessor for IGA. The storage format is DWG or DXF, which consist of ASCII data. From these, the curve data can then be easily extracted. For area data, it is more difficult. In AutoCAD, there are two options for modelling curves, either by identifying the curve with data points or by modelling the curve with control points. It is also possible to add control points, modify the degree of the polynomial, modify the position of the control points, or weight parameters.

#### 2.1.4. Blender

A comprehensive environment for modelling, animation, and rendering. Geometrically, Blender provides excellent support for B-REP modelling with polygonal mesh. Although Blender has a few NURBS tools, it is made up of straight edges. This is especially reflected when creating circles or curves. It needs to be part of the preprocessor chain and not the whole of it [2].

#### 2.1.5. MATLAB and GNU Octave

Their advantage is easy orientation in the notation. The disadvantage is that NURBS is not part of these programs. You have to define them or use the available toolboxes for extensions. Another disadvantage is that these programs do not provide the interactive display capabilities that Rhinoceros 3D or Blender provide.

The open-source package GeoPDE can now be used. It is very user friendly, contains many examples and detailed step-by-step descriptions. Thanks to it, MATLAB and GNU Octave do not remain only preprocessors, but it is possible to perform IGA directly in it. The IgaFEM package is still usable for MATLAB.

## 3. Interpolation Using B-Splines and NURBS Method

NURBS represents a special case of a B-spline curve where the nodal vector is “non-uniform”. “Non-uniformity” allows to plot curves with arbitrary degrees of continuity [9]. They are significant precisely for this ability, which allows them to represent every object quite accurately. This is possible due to the projective transformation over parts of a quadratic curve.

For different orders of continuity between curves, the following can be said:

$C^{-1}$ curve is not continuous,$C^{0}$ curve is continuous,$C^{1}$ retains continuity even after the first derivative,$C^{2}$ is continuous even after the second derivative,$C^{n}$ continuous up to the nth derivative.

It is true that any node can be a breakpoint of several polynomial curves. That is, a single polynomial is expressed using two non-repeating nodes. If two of the three control points are to be “fixed” at the ends of the curve, then the first and last nodes must be repeated p+1 times. This also achieves discontinuity at the endpoints.

By inserting a node into an interval of control points that is from −1 to 1, one can increase the number of basic functions and hence also increase the number of control points required to plot a B-spline curve without increasing the degree of the polynomial of the curve [3].

For the NURBS curve, control points are obtained using a projective transformation. Variables such as the projective curve $C^{w}\left(\xi \right)$, are related to the B-splines, which is the projection of the control points, $B_{i}^{w}$. For NURBS, it is equally true that C(ξ) relates to the projective curve and $B_{i}$ relates to the projection of points. The control points for the NURBS curve are obtained from the relations [1]
(1)$${\left(B_{i}\right)}_{j}=\frac{{\left(B_{i}^{w}\right)}_{j}}{w_{i}};\;\;\;\;j=1,\ldots ,d,$$
(2)$$w_{i}={\left(B_{i}^{w}\right)}_{d+1},$$
where ${\left(B_{i}\right)}_{j}$ is the *j*-th component of the vector $B_{i}$, $w_{i}$ is the *i*-th weighting parameter. Redistributing the projected control points by the weights is equivalent to using a projective transformation. It is necessary to apply this procedure to each point of the curve in order to define the weighting function
(3)$$W\left(\xi \right)=\overset{n}{\underset{i=1}{\sum }}N_{i,p}\left(\xi \right)w_{i},$$
where $N_{i,p}\left(\xi \right)$ is the standard B-spline basis function. Consequently, the NURBS curve can be defined as
(4)$${\left(C\left(\xi \right)\right)}_{j}=\frac{{\left(C^{w}\left(\xi \right)\right)}_{j}}{W\left(\xi \right)};\;\;\;\;j=1,\ldots ,d,$$
where $C^{w}\left(\xi \right)$ and W(ξ) are both piecewise polynomial functions. The NURBS curve C(ξ) is a piecewise rational function, with a polynomial partitioned by another polynomial in each element. In NURBS, the two polynomials have the same degree.

If a B-spline curve has points with $C^{0}$ continuity and it is necessary to express this state as a quadratic function, then the values of the nodes at these positions must be multiplied by two. Often, however, the level of continuity is limited by the shape of the projective curve rather than by the curve itself. To give an idea, there is no solution to constructing a circle without nodes where the continuity is at the $C^{0}$ level. To perform the affine transformation of a NURBS object, the affine transformation must be applied directly to the control points, but the weighting parameters are preserved. Although the weight parameters are bound to a specific control point, they are not a component of it, which is a common mistake when constructing a NURBS data structure.

The algebraic approach allows to understand the construction of NURBS objects. Similar to B-splines, the weighting function is a scalar, piecewise polynomial function for the (d+1) component of the projected curve.
(5)$$R_{i}^{p}\left(\xi \right)=\frac{N_{i,p}\left(\xi \right)w_{i}}{W\left(\xi \right)}=\frac{N_{i,p}\left(\xi \right)w_{i}}{\sum_{\hat{i}=1}^{n}N_{\hat{i},p}\left(\xi \right)w_{\hat{i}}},$$
which is a piecewise rational function [1]. Using this equation with control points, one can arrive at the NURBS curve equation [1]
(6)$$C\left(\xi \right)=\overset{n}{\underset{i=1}{\sum }}R_{i}^{p}\left(\xi \right)B_{i}.$$

### 3.1. NURBS—Basis for Analysis

In IGA, a basis is selected that is capable of accurately representing the geometry, which is then used as the basis for the fields to be approximated.

Sufficient conditions for a basic proof of convergence are provided for a wide range of problems. However, the following must hold:*C*^1^ continuity for the interior of the element,*C*^0^ continuity at the boundaries of the element,completeness—on any element is a basis able of representing all linear functions.

These conditions are necessary to ensure convergence for the different elements.

### 3.2. Numerical Methods

There are several methods that have found their application in IGA. The most widely used is Galerkin finite element analysis. Others worth mentioning are the collocation method, the least squares method, and mesh-free methods.

#### Algorithms for Systematic Modification of NURBS

Most commonly, 3 basic algorithms are used to increase the flexibility of objects created from NURBS, while modifying the properties associated with the basic functions. Their names are:knot insertion, knot refinement,increase in order, degree of rotation,k-refinement.

These modifications not only affect the geometry but can be used to obtain more accurate values of displacements, relative strains and stresses.

### 3.3. Elastostatics in the IGA

Here, $u_{i}$ represents the components of the displacement vector and the Cauchy stress tensor σ has Cartesian components $\sigma_{ij}$. The strain tensor ε  with components $\varepsilon_{ij}$ is determined to be the symmetric part of the displacement gradient
(7)$$\varepsilon_{ij}=u_{\left(i,j\right)}\equiv \frac{u_{i,j}+u_{j,i}}{2}.$$

Next, the generalized Hooke’s law can be introduced
(8)$$\sigma_{ij}=c_{ijkl}\varepsilon_{kl},$$
where $c_{ijkl}$ represent the elastic coefficients, which have two characteristic properties:symmetry,positive definiteness.

The material is expected to be homogeneous and isotropic. Thus, the elastic coefficients take the form
(9)$$c_{ijkl}=\lambda \delta_{ij}\delta_{kl}+\mu \left(\delta_{ik}\delta_{jl}+\delta_{il}\delta_{jk}\right),$$
where λ and μ represent the Lamé parameters and $\delta_{ij}$ is the Kronecker delta.

### 3.4. Strong Formulation

Consequently, a strong formulation must be defined. It is given that fi:Ω→ℝ,  gi:ΓDi and hi:ΓNi→ℝ and it is necessary to find ui: Ω ¯→ℝ such that
(10)σij,j+fi=0 in  Ω,
(11)ui=gi in ΓDi,
(12)σijnj=hi in ΓNi.

The last two Equations (11) and (12) are generalizations of the Dirichlet and Neumann boundary conditions. These conditions are utilized in each direction independently, so that $\bar{\Gamma_{D_{i}}⋃\Gamma_{N_{i}}}=\Gamma$ and $\Gamma_{D_{i}}⋂\Gamma_{N_{i}}=Ø$ for i=1,…, d. $G_{i}$ and $h_{i}$ are the prescribed displacements.

### 3.5. Weak Formulation

The test solution space $S_{i}$ and the weight space $V_{i}$ are introduced. In addition to the existing conditions, one more condition is added. Each $w_{i}\in V_{i}$ satisfies $w_{i}=0$ na $\Gamma_{D_{i}}$. Equation (10) is multiplied by the weight function and integrated piecewise. The goal is to obtain the variational form. The purpose is to find $u_{i}\in S_{i}$ such that for all $w_{i}\in V_{i}$ one can write
(13)$$\underset{\Omega }{\int }w_{\left(i,j\right)}\sigma_{ij}d\Omega =\underset{\Omega }{\int }w_{i}f_{i}d\Omega +\overset{d}{\underset{i=1}{\sum }}\left(\underset{\Gamma_{N_{i}}}{\int }w_{i}h_{i}d\Gamma \right)$$

This equation can also be expressed more concisely when $S=\left\{u|u_{i}\epsilon S_{i}\right\}$ and let $V=\left\{w|w_{i}\epsilon V_{i}\right\}$. Given $f=\left\{f_{i}\right\}$, $g=\left\{g_{i}\right\}$ and $h=\left\{h_{i}\right\}$. The search is for u ∈ S  such that for all w ∈ V
(14)a(w,u)=L(w)
(15)$$a\left(w,u\right)=\underset{\Omega }{\int }w_{\left(i,j\right)}c_{ijkl}u_{\left(k,l\right)}d\Omega ,$$
(16)$$L\left(w\right)=\underset{\Omega }{\int }w_{i}f_{i}d\Omega +\overset{d}{\underset{i=1}{\sum }}\left(\underset{\Gamma_{N_{i}}}{\int }w_{i}h_{i}d\Gamma \right).$$

### 3.6. Galerkin’s Method

In order to transform the weak formulation of the problem into a system of algebraic equations, it is necessary to apply Galerkin’s method, which consists of constructing finite-dimensional approximations and working in the subspaces $S^{h}\in S$ and $V^{h}\in V$ Briefly, these are subsets that will be associated with the bridged space of the isoparametric basis. NURBS bases are used to define these subspaces, and they contain vector control variables. If $g^{h}\in$ $S^{h}$ are determined, where $g_{i}^{h}|\;\Gamma_{D_{i}}=g_{i}$, then for all $u^{h}\in \;S^{h}$
(17)$$u^{h}=v^{h}+g^{h}.$$
where $v^{h}\in \;V^{h}$. For the Galerkin approximation, when applied to the weak form, we use the relation
(18)$$a\left(w^{h},v^{h}\right)=L\left(w^{h}\right)-a\left(w^{h},g^{h}\right).$$

It is possible to achieve an increase in accuracy by defining $\eta =\left\{1,\ldots ,n_{np}\right\}$ indices of all NURBS functions that define the geometry. Respectively, let $\eta_{g_{i}}\subset \eta$ contain the indices of all basis functions that are nonzero on $\Gamma_{D_{i}}$. Then, for the i-th component $u^{h}\in \;S^{h}$ can be used in the equation
(19)$$u_{i}^{h}=\underset{A\in \eta -\eta_{g_{i}}}{\sum }N_{A}d_{iA}+\underset{B\in \eta_{g_{i}}}{\sum }N_{B}g_{iB}=\underset{A\in \eta -\eta_{g_{i}}}{\sum }N_{A}d_{iA}+g_{i}^{h}.$$

The equation is a simple vector generalization, with $d_{iA}$ representing the i-th component of the control variable $d_{A}$. A similar formulation can be written for $w^{h}\in V^{h}$
(20)$$w_{i}^{h}=\sum_{A\in \eta -\eta_{g_{i}}}N_{A}c_{iA}.$$

Finally, it is possible to express $u^{h}$ and $w^{h}$ using
(21)$$u^{h}=u_{i}^{h}e_{i}\;\;\;a\;\;w^{h}=w_{i}^{h}e_{i},$$
where $e_{i}$ in three-dimensional space can be expressed by the relations
(22)$$e_{1}=\left(\begin{array}{c} 1\\ 0\\ 0 \end{array}\right)\;\;e_{1}=\left(\begin{array}{c} 0\\ 1\\ 0 \end{array}\right)\;\;e_{1}=\left(\begin{array}{c} 0\\ 0\\ 1 \end{array}\right).$$

The purpose of all this was to obtain equations to express the matrix formulation of the problem
(23)Kd=F,
where
(24)$$K=\left[K_{PQ}\right],\;d=\left\{d_{Q}\right\},\;F=\left\{F_{P}\right\},$$
whereby the IEN connectivity field must be used to assign a local function number and element number to the global basis function. One equation was assigned to each such global function. Then there are d equations for each global function. Next, a second degree of connectivity must be introduced using the ID field, which is related to the degrees of freedom i=1,…,d and the number of the global function $A\in \eta -\eta_{g_{i}}$ and returns the equation number P=ID(i,A). With this knowledge, the number of the global equations can be obtained by constructing ID and IEN to obtain P=ID(i,IEN(a,e)). For use in three-dimensional space, a connectivity field *LM* is introduced which includes both of the mentioned fields. Then P=LM(i,a,e)=ID(i,IEN(a,e)). Given Equation (24), it is necessary to write that Q=ID(j,B).
(25)$$K_{PQ}=a\left(N_{A}e_{i},N_{B}e_{j}\right),$$
(26)$$F_{P}=L\left(N_{A}e_{i}\right),$$
and
(27)$$d_{Q}=d_{jB}.$$

### 3.7. Construction of Global and Local Stiffness Matrices

First, it is necessary to define a vector of strains, then a simplified form of the fourth-order elastic coefficient tensor can be introduced into the matrix D, where $D_{IJ}=c_{ij\;kl}$. With this knowledge, the stress vector can be defined as
(28)σ=Dε.

This is often referred to as Voigt notation. Equation (15) is then used, which is then translated into the form
(29)$$a\left(w,u\right)=\int \varepsilon {\left(w\right)}^{T}D\varepsilon \left(u\right)d\Omega ,$$
then the notation is expanded
(30)$$\varepsilon \left(N_{A}e_{i}\right)=B_{A}e_{i},$$
where
(31)$$B_{A}=\left[\begin{array}{ccc} N_{A,1} & 0 & 0\\ 0 & N_{A,2} & 0\\ 0 & 0 & N_{A,2}\\ 0 & N_{A,3} & N_{A,2}\\ N_{A,3} & 0 & N_{A,1}\\ N_{A,2} & N_{A,1} & 0 \end{array}\right].$$

With this knowledge, the global stiffness matrix is rewritten in the form
(32)$$K_{PQ}=e_{i}^{T}\underset{\Omega }{\int }B_{A}^{T}DB_{B}d\Omega e_{j}.$$

By using all the elements and creating local stiffness matrices and force vectors, a global stiffness matrix and force vector are created. With d spatial dimension and local shape function $n_{en}$, the input values on the local stiffness matrix on element $\Omega^{e}$ are calculated as
(33)$$k_{pq}^{e}=e_{i}^{T}\underset{\Omega^{e}}{\int }B_{a}^{T}DB_{b}d\Omega e_{j},$$
while p=d(a−1)+i and q=d(b−1)+j. Similarly, the terms of the local force vector can be expressed
(34)$$f_{p}^{e}=\underset{\Omega^{e}}{\int }N_{a}f_{i}d\Omega +\underset{\Gamma_{h_{i}}^{e}}{\int }N_{a}h_{i}d\Omega -\overset{d\cdot n_{em}}{\underset{q=1}{\sum }}k_{pq}^{e}g_{q}^{e}.$$
where $\Gamma_{h_{i}}^{e}$ is the edge intersection of the element with $\Gamma_{h_{i}}$ and $g_{q}^{e}=g_{Q}$. In practice, Gaussian quadrature is used to compute integrals.

The relations given in this section, except where otherwise noted, are taken from the literature [37].

## 4. Using GeoPDE in MATLAB at IGA

This subchapter presents the results obtained using IGA in MATLAB with the help of the GeoPDE package. Several thin-walled structural elements were created in MATLAB and in parallel in a program using FEM. The aim is to compare the result obtained from IGA and FEA for solution of the same structures. Symmetry could not be used in many cases, since GeoPDE does not allow it for most solvers. The finite element computations were accomplished by the program Ansys using SHELL181 elements. SHELL181 elements are suitable for the analysis of shell structures. SHELL181 is a four-node element with six degrees of freedom at each node (translations in the three directions, and rotations about three directions).

### 4.1. Model of Hemispherical Element

The selected model of the hatch element has a radius of 10 mm and a wall thickness of 0.25 mm. An isotropic material (structural steel) is assumed. The definition of the boundary conditions for the FEM is shown in Figure 1a. The model was loaded with an internal pressure of 1.2 MPa.

Figure 1b presents the definition of parameters, material constants and boundary conditions in MATLAB. By simulation, we obtained the displacement field in the load direction and the total displacements by both solution methods. The displacement field is shown in Figure 2a,b and Figure 3a,b. Figure 4 is a plot showing a comparison of the displacements obtained by the FEM and IGA simulations, which shows an almost perfect agreement of the results obtained. The computed displacements for the selected points of the structure are given in Table 1.

### 4.2. Funnel

The next modelled object was a funnel. The edge where the liquid would leak out was chosen as the point of interest for displacements. The type of model was chosen with respect to tanks whose bottoms may consist of such features. Figure 5a shows the boundary conditions specified in FEA and Figure 5b shows the programming procedure for determining the geometric and boundary conditions in MATLAB. By simulating both methods, the displacements in the *x*-axis direction due to load, Figure 6, and the total displacements due to load, Figure 7, were monitored. Figure 8 is a plot showing the comparison of the results obtained for the IGA and in FEA for the 0.5 mm and 0.2 mm meshes, respectively. The computed displacements for the selected points of the structure are given in Table 2. The compared results provide almost identical agreement.

### 4.3. Scordelis–Lo Roof

This is a classical problem addressed in a number of papers. The only difference is that the original material is replaced with structural steel. The load is applied in one direction from the outside towards the axis of rotation.

Symmetry was exploited in the FEM analysis. Figure 9a shows the boundary conditions on the symmetric part of the model. Figure 9b presents the procedure of programming the boundary conditions in MATLAB. Figure 10 and Figure 11 show the results of displacements in the load direction and total displacements, respectively.

Figure 12 presents a plot comparing the displacement results obtained by the two methods in the *z*-axis direction. The computed displacements for the selected points of the structure are given in Table 3. Even with the refinement of the finite element mesh in the FEM analysis, we obtained an almost complete agreement in the results with the IGA method.

### 4.4. Circular Plate

The circular plate is fixed around the perimeter, so all degrees of freedom of movement are taken away at the edge of the plate. The load is applied in a direction perpendicular to the face of the plate.

Figure 13a shows the boundary conditions of the model. Figure 13b shows the procedure for programming the boundary conditions in MATLAB.

By simulating the loading of the circular plate, displacement results were obtained. Figure 14a shows the total displacement fields obtained in FEM analyses and Figure 14b shows the IGA methods in MATLAB.

For the circular plate, an analytical solution was accomplished for the control. The deflection of a circular plate, fixed at the edge and loaded in compression over the surface, can be calculated from the relation
(35)$$w\left(r\right)=\frac{p}{64D}\left(R^{2}-r^{2}\right),$$
where
D is a plate stiffness. For given material and dimensions is
D=146,520.1 Nmm.

The computed displacements for the selected points of the structure are given in Table 4. Figure 15 shows a comparison of the displacement results obtained by each method.

## 5. Conclusions

The aim of the study was to perform isogeometric analysis on thin-walled elements of various shapes, but mostly on elements with at least one radius, and to evaluate its accuracy in comparison with the finite element method. This is a complex problem involving the issue of geometry interpolation using NURBS and the formulation of appropriate computational procedures. The description of isogeometric analysis itself consists of the Galerkin method, the NURBS refinement algorithm, the output of isogeometric analysis, and the definition of the relations valid for elastostatics in isogeometric analysis.

Isogeometric analysis has the following main advantages with respect to the finite element method:There is no geometric approximation error, due to the fact the geometry of the analyzed body is represented exactly.Interpolation of geometry can be obtained directly from a computer-aided design tool.Wave propagation problems are better described due to the reduction of numerical dispersion and dissipation errors.

In principle, there is only one disadvantage of this method compared to the classical finite element method and that is a more complicated theory.

The work clearly demonstrated that for thin-walled structural elements, the isogeometric analysis gives similar results to the finite element method. A number of thin-walled structural problems in elastostatics were solved. In all cases, a high agreement was shown even in cases where refinement of the mesh occurred in the finite element method.

Table 5 shows the mesh parameters in each solved case along with the computation times. In this case, however, these times are of limited informative value because different solvers are used in the FEM calculations and in the isogeometric analysis, moreover, they are programmed in programming languages of different performance.

## Figures and Tables

**Figure 1 materials-15-06516-f001:**
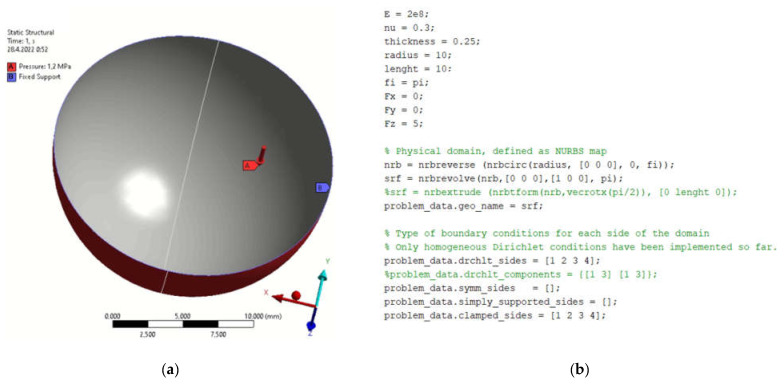
Defining parameters, material constants and boundary conditions in (**a**) FEM program; (**b**) MATLAB.

**Figure 2 materials-15-06516-f002:**
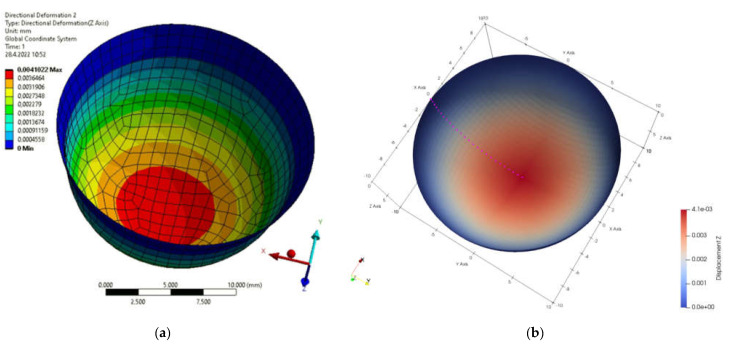
Displacements in load direction (**a**) FEM program; (**b**) MATLAB.

**Figure 3 materials-15-06516-f003:**
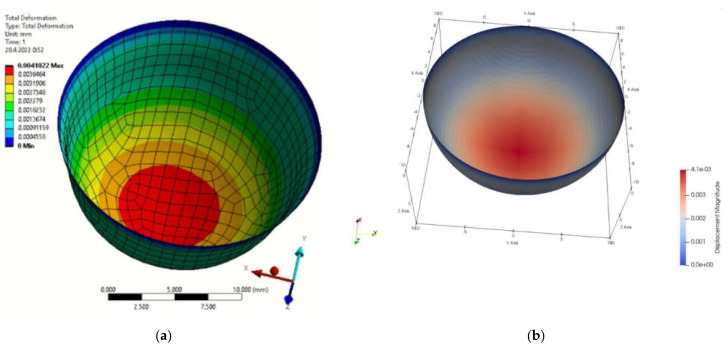
Total displacement field (**a**) FEM program; (**b**) MATLAB.

**Figure 4 materials-15-06516-f004:**
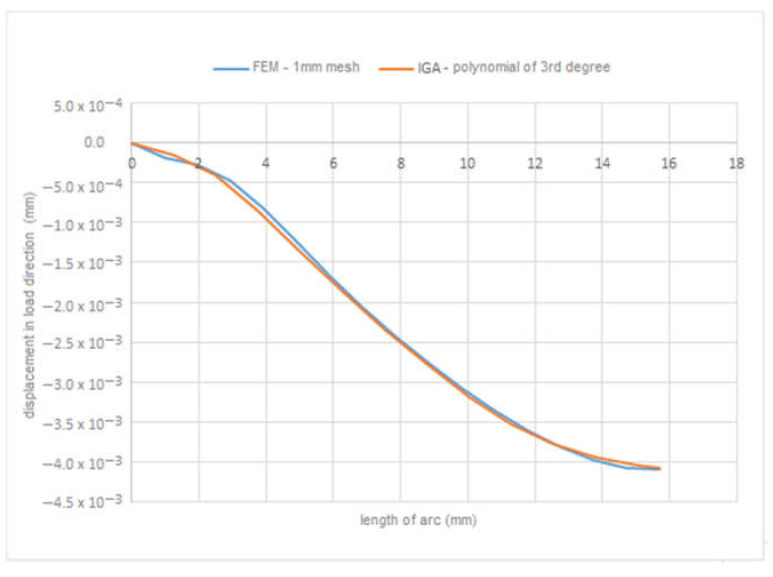
Comparison of displacements in load direction.

**Figure 5 materials-15-06516-f005:**
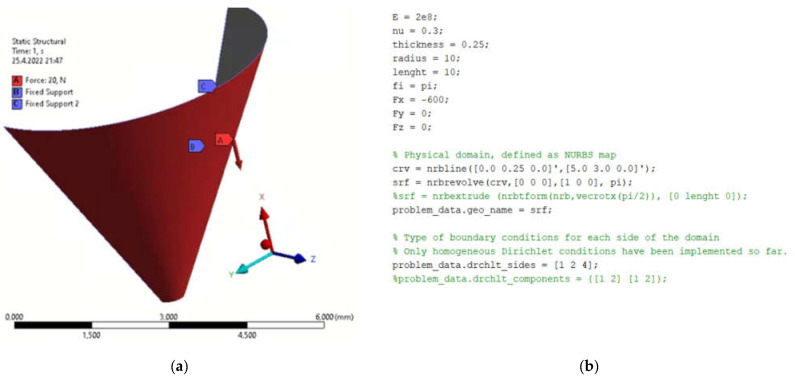
Geometric characteristics, boundary conditions and material properties (**a**) FEM program; (**b**) MATLAB.

**Figure 6 materials-15-06516-f006:**
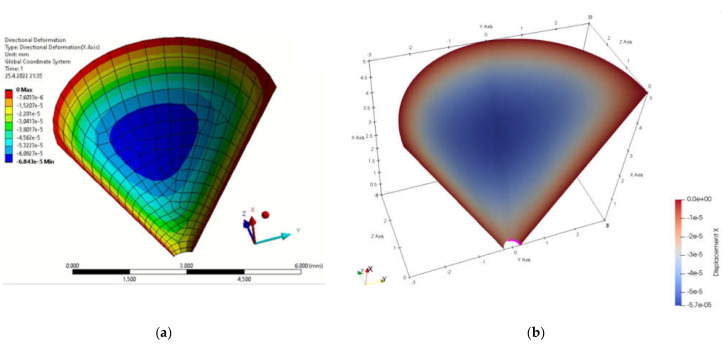
Displacement due to load in x direction (**a**) FEM program; (**b**) MATLAB.

**Figure 7 materials-15-06516-f007:**
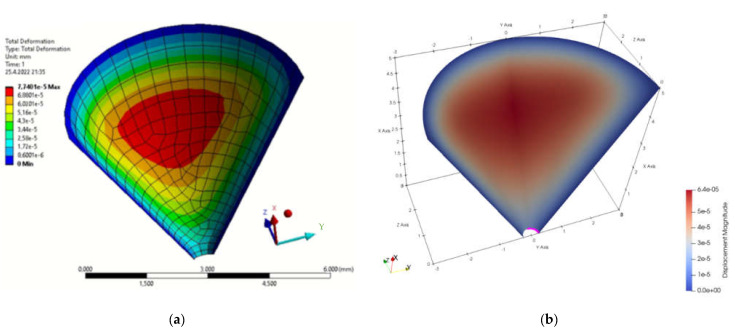
Total displacement field (**a**) FEM program; (**b**) MATLAB.

**Figure 8 materials-15-06516-f008:**
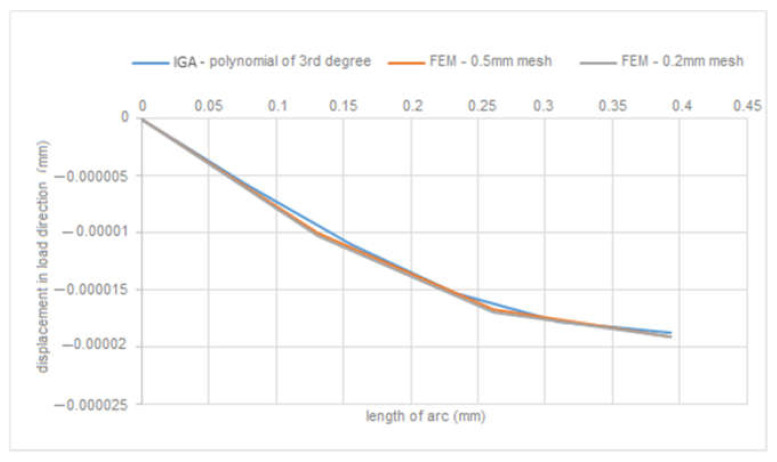
Comparison of obtained displacements.

**Figure 9 materials-15-06516-f009:**
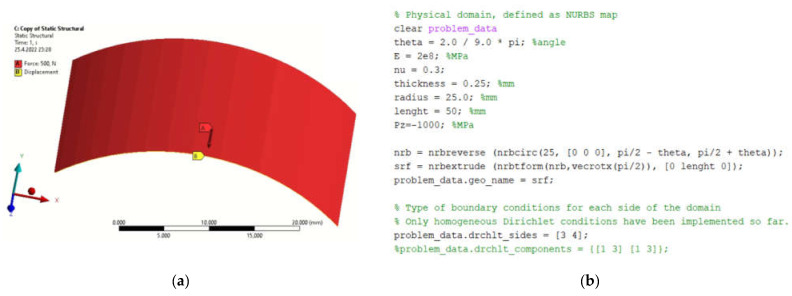
Geometric characteristics, boundary conditions and material properties (**a**) FEM program; (**b**) MATLAB.

**Figure 10 materials-15-06516-f010:**
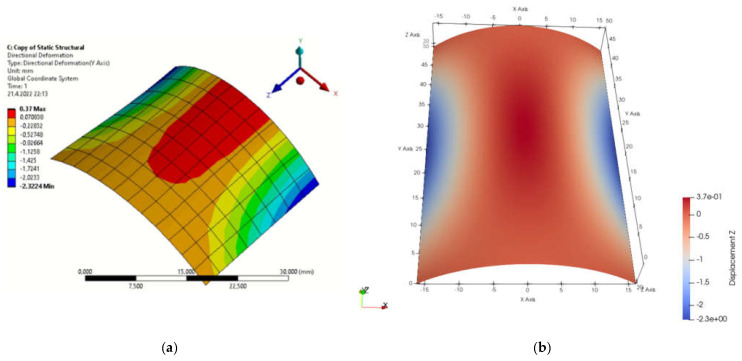
Displacements in load direction (**a**) FEM program; (**b**) MATLAB.

**Figure 11 materials-15-06516-f011:**
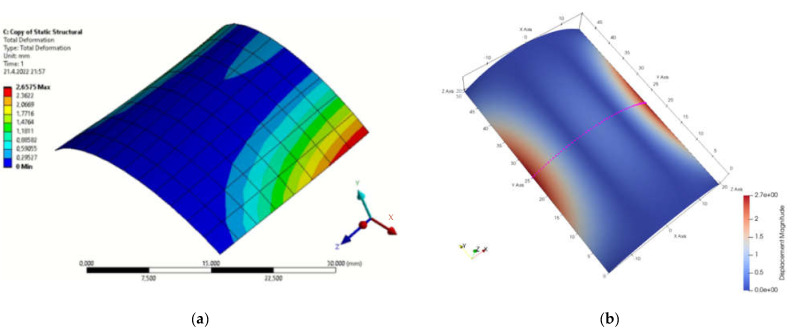
Total displacement field (**a**) FEM program; (**b**) MATLAB.

**Figure 12 materials-15-06516-f012:**
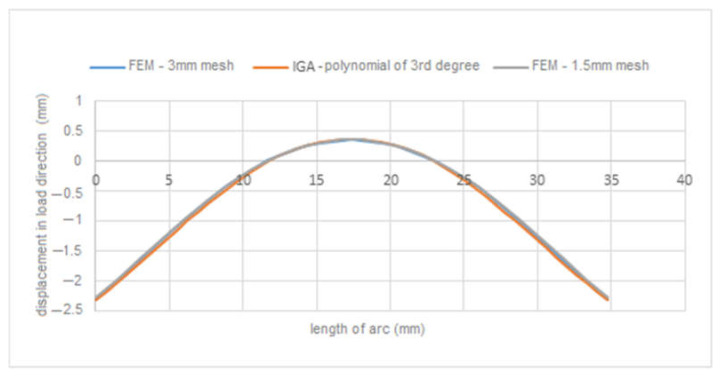
Comparison of displacements in the *z*-axis direction.

**Figure 13 materials-15-06516-f013:**
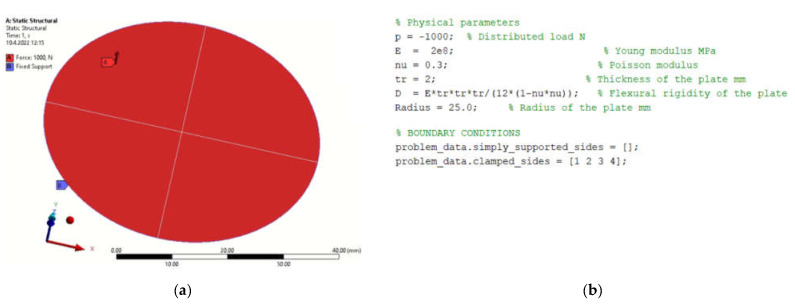
Geometric characteristics, boundary conditions and material properties (**a**) FEM program; (**b**) MATLAB.

**Figure 14 materials-15-06516-f014:**
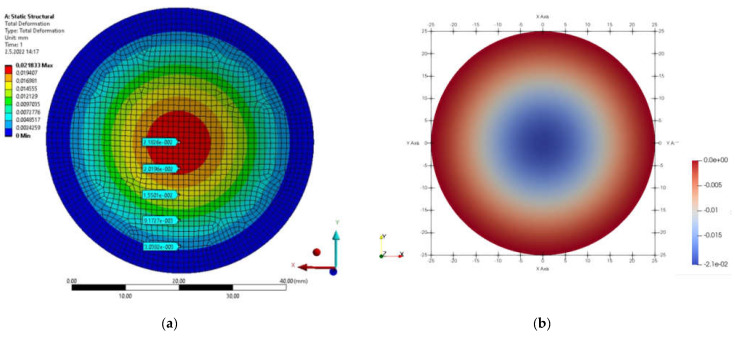
Total displacement field (**a**) FEM program; (**b**) MATLAB.

**Figure 15 materials-15-06516-f015:**
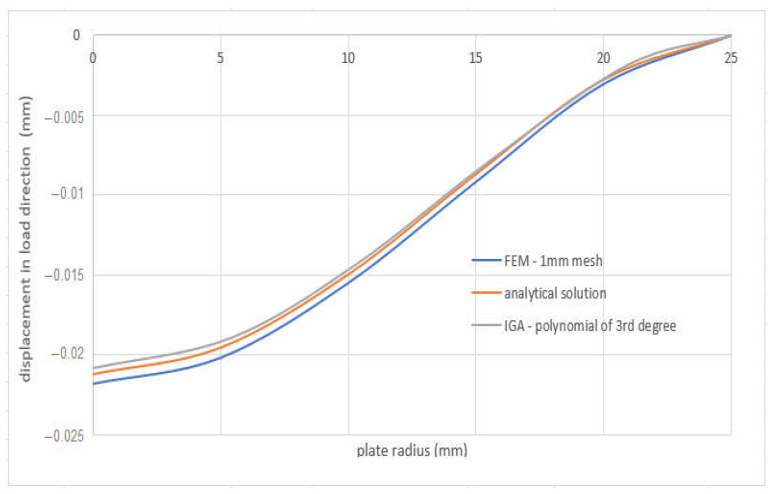
Comparison of result displacements of FEA and IGA simulation.

**Table 1 materials-15-06516-t001:** Computed displacements of the points for the different solution methods.

	FEM—1 mm Mesh	IGA—Polynomial of 3rd Degree	Absolute Value of Displacement Differences
Length of Arc (mm)	*w*(*r*) (mm)	*w*(*r*) (mm)	
0	0	0	0
1.25	−0.000185	−0.00016	0.00003
5.02	−0.00129	−0.00137	0.00008
8.79	−0.00277	−0.00279	0.00002
10.04	−0.00398	−0.00319	0.00079
13.81	−0.00398	−0.00394	0.00004
15.7	−0.0041	−0.00409	0.00001

**Table 2 materials-15-06516-t002:** Computed displacements of the points for the different solution methods.

	FEM—0.2 mm Mesh	FEM— 0.5 mm Mesh	IGA—Polynomial of 3rd Degree	Absolute Value of Displacement Differences between IGA and FEM—0.2 mm mesh	Absolute Value of Displacement Differences between IGA and FEM—0.5 mm mesh
Length of Arc (mm)	*w*(*r*) (mm)	*w*(*r*) (mm)	*w*(*r*) (mm)		
0.13	−0.0000102	−0.0000101	−0.0000092	0.000001	0.0000009
0.26	−0.0000170	−0.0000168	−0.0000158	0.0000012	0.0000001
0.39	−0.0000192	0.0000191	−0.0000188	0.0000004	0.0000379

**Table 3 materials-15-06516-t003:** Computed displacements of the points for the different solution methods.

	FEM—1.5 mm Mesh	FEM—3 mm Mesh	IGA—Polynomial of 3rd Degree	Absolute Value of Displacement Differences between IGA and FEM—1.5 mm Mesh	Absolute Value of Displacement Differences between IGA and FEM—3 mm Mesh
Length of Arc (mm)	*w*(*r*) (mm)	*w*(*r*) (mm)	*w*(*r*) (mm)		
0.00	−2.2865	−2.3209	−2.3209	0.0344	0
2.89	−1.6750	−1.6760	−1.6760	0.001	0
5.79	−1.0273	−1.0273	−1.0276	0.0003	0.0003
8.68	−0.4421	−0.4421	−0.4421	0	0
11.58	0.0689	0.0689	0.0689	0	0
14.47	0.2800	0.2800	0.2800	0	0
17.36	0.3580	0.3700	0.3700	0.012	0
20.26	0.2785	0.2786	0.2786	0.0001	0
23.15	0.0040	0.0040	0.0040	0	0
26.04	−0.4490	−0.4462	−0.4421	0.0069	0.0041
28.94	−0.9696	−1.0324	−1.0324	0.0628	0
31.83	−1.6337	−1.6818	−1.6818	0.0481	0
34.71	−2.2860	−2.3224	−2.3224	0.0364	0

**Table 4 materials-15-06516-t004:** Computed displacements of the points for the different solution methods.

FEM—1mm Mesh		IGA—Polynomial of 3rd Degree		Analytical Solution		Absolute Value of Displacement Differences between FEM and Analytical Solution (mm)	Absolute Value of Displacement Differences between IGA and Analytical Solution (mm)
***r*** (mm)	*w*(*r*) (mm)	***r*** (mm)	*w*(*r*) (mm)	***r*** (mm)	*w*(*r*) (mm)		
0	−0.021826	0	−0.0208103	0	−0.0212	0.00063	0.00039
5	−0.020196	5.39	−0.01893	5	−0.01954	0.00066	0.00061
10	−0.01551	10.28	−0.01435	10	−0.01496	0.00055	0.00061
15	−0.0091727	15.577	−0.0078	15	−0.00868	0.00049	0.00088
20	−0.0030382	20.9	−0.00188	20	−0.00275	0.00029	0.00087
25	0	25	0	25	0	0	0

**Table 5 materials-15-06516-t005:** Comparison of mesh parameters and solution times.

	IGA	FEM	
**Hemisphere**	**Polynom of the 3rd Degree**	**Mesh Density 1 mm**	**Mesh Density 0.7 mm**
Time (s)	3.1506	1.938	2.203
Number of nodes	1369	694	1378
Number of elements	-	666	1336
**Funnel**	**Polynom of the 3rd degree**	**Mesh density 0.5 mm**	**Mesh density 0.2 mm**
Time (s)	1.878	1.828	2.016
Number of nodes	703	276	982
Number of elements	576	246	927
**Scordelis–Lo roof**	**Polynom of the 3rd degree**	**Mesh density 3 mm**	**Mesh density 1.5 mm**
Time (s)	0.9635	1.719	2.062
Number of nodes	361	234	816
Number of elements	256	204	759
**Circular plate**	**Polynom of the 3rd degree**	**Mesh density** **1 mm**	
Time (s)	0.157	2.328	
Number of nodes	144	2519	
Number of elements	-	2461	

In the future, it would be possible to extend this work to other solvers, whether in MATLAB (IgaFEM), and others such as feappv, or Rhinoceros 3D. Next steps could be the isogeometric analysis of solids formed from NURBS “patches”, fluid flow, etc.

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
