# Peer review of "Comparison of Structural Analysis of Thin-Walled Structures Accomplished by Isogeometric Analysis and the Finite Element Method"

_materials, 2022, doi:10.3390/ma15196516_

Round 1
Reviewer 1 Report
This paper presents a comparison between IGA and finite element methods for structural analysis of thin-walled structures using elasticity equations. Both methods where implemented in Matlab for objects modelled using a single patch. The paper is well organized and it deserves publication in Materials after the following revisions:
1- The solution of the linear system (23) should be well discussed and it numerical solution in both methods should be assessed in the manuscript. The condition number of the associated matrices for both method should be provided and to have a fair comparison between the two methods, the same solver needs to be used for the linear systems in both method. Authors are requested to provide, at least for one of the considered examples, a table with condition number and CPU times for solving the linear systems in each method.
2- The results presented in section 4 lack a serious assessment of the performance of the proposed method as no computational costs are given for these results. Authors should compare results of accuracy and efficiency for the methods using several mesh refinements.
3- For a good assessment of the proposed techniques, results obtained from IGA method should be compared to those obtained from the FEM method for an example with known exact solution or manufactured analytical solution to show the errors and the convergence rates for each method.
Reviewer 2 Report
The authors provide a systematic derivation on employing the isogeometric analysis for solving thin-walled structures (like plates and shells), which can be significantly important as these structs are seen in aerospace, automotive and other advanced fields. Nevertheless, the authors fail to provide enough discussion on the advantages of the IGA method over other numerical methods. In addition, the current paper seems more of a theoretical work, rather than an engineering paper. I would suggest the author to re-submit their manuscript after a major revision addressing the following concerns:
Line 27:temperature is a scalar, can the authors describe why they consider it as a tensor quantity?
Please provide a reference (or references) for eqns. appear in chp.3.
Chp. 4.1: What is the FEM program used in this study? Seems like Abaqus, but the authors did not explicitly claimed.
Chp. 4.1: When comparing the results from FEM and IGA, a basic description of the element type employed (of FEM), mesh quality (are the two simulations the same?), and similar conditions should be clarified in advance.
Figure 4: In addition to read the curves by eyes, is there any other way the authors could use to show the difference/similarity between results generated by the two methods? (i.e. statistically?)
It seems that the other sections in Chapter 4 suffer similar issues as I mentioned above… More explanation, analysis and discussion are needed.
BIGGEST concern: The authors mentioned in the abstract that IGA method is superior mainly due to it cuts off the simulation time (computationally effective?). However, I did not see there was any discussion or analysis presenting that IGA method is better in saving computational time. Please justify this.
Author Response
We thank the reviewers for their suggestions to improve the paper. It helped us to re-think the formulations we used and make them better. We have made substantial changes to the manuscript, taking all comments into account.
We found that in the References list was missing a number at item [19], so the numbering from this number had to be adjusted. However, in the text the numbering corresponded to the actual modified numbering. In addition, we have added entry [37] to the reference list.
The authors provide a systematic derivation on employing the isogeometric analysis for solving thin-walled structures (like plates and shells), which can be significantly important as these structs are seen in aerospace, automotive and other advanced fields. Nevertheless, the authors fail to provide enough discussion on the advantages of the IGA method over other numerical methods. In addition, the current paper seems more of a theoretical work, rather than an engineering paper. I would suggest the author to re-submit their manuscript after a major revision addressing the following concerns:
Reply:
Some sentences regarding advantages and disadvantages of isogeometic analysis are included into Conclusions.
Line 27:temperature is a scalar, can the authors describe why they consider it as a tensor quantity?
Reply:
Since in the text we mentioned quantities of different nature (displacement / vector (first-order tensor), temperature / scalar (zero-order tensor), and here may also be included the quantities stress and strain / second-order tensors) we have used the general term tensor to characterize all these quantities.
Please provide a reference (or references) for eqns. appear in chp.3.
Reply:
References to the equations have been added to Section 3.
Chp. 4.1: What is the FEM program used in this study? Seems like Abaqus, but the authors did not explicitly claimed.
Reply:
The finite element computations were accomplished by program Ansys.
Chp. 4.1: When comparing the results from FEM and IGA, a basic description of the element type employed (of FEM), mesh quality (are the two simulations the same?), and similar conditions should be clarified in advance.
Reply:
We have added the element type used in FE analysis and mesh quality is described in the last table – Table 5, added int revision text.
Figure 4: In addition to read the curves by eyes, is there any other way the authors could use to show the difference/similarity between results generated by the two methods? (i.e. statistically?)
It seems that the other sections in Chapter 4 suffer similar issues as I mentioned above… More explanation, analysis and discussion are needed
Reply:
We have added new tables consisting of data connected with computed displacements at some selected points.
BIGGEST concern: The authors mentioned in the abstract that IGA method is superior mainly due to it cuts off the simulation time (computationally effective?). However, I did not see there was any discussion or analysis presenting that IGA method is better in saving computational time. Please justify this.
Reply:
The conclusion of the paper was exceeded by several considerations.
Round 2
Reviewer 1 Report
The authors have managed to address my comments made on the first version of their manuscript. I am satisfied for the answers provided and I do suggest acceptance of this manuscript.
Reviewer 2 Report
I appreciate the authors' efforts on addressing my concerns. The current version can be accepted.